# Antioxidant Capacity and Profiles of Phenolic Acids in Various Genotypes of Purple Wheat

**DOI:** 10.3390/foods11162515

**Published:** 2022-08-20

**Authors:** Vladimir P. Shamanin, Zeynep H. Tekin-Cakmak, Elena I. Gordeeva, Salih Karasu, Inna Pototskaya, Alexandr S. Chursin, Violetta E. Pozherukova, Gorkem Ozulku, Alexey I. Morgounov, Osman Sagdic, Hamit Koksel

**Affiliations:** 1Department of Agronomy, Breeding and Seed Production of the Agrotechnological Faculty, Omsk State Agrarian University, 1 Institutskaya pl., 644008 Omsk, Russia; 2Department of Nutrition and Dietetics, Health Sciences Faculty, Istinye University, 34010 İstanbul, Turkey; 3Department of Functional Genetics of Cereals, Institute of Cytology and Genetics SB RAS, Prospect Akademika Lavrent’yeva, 10, 630090 Novosibirsk, Russia; 4Department of Food Engineering, Faculty of Chemical and Metallurgical Engineering, Yildiz Technical University, 34349 Istanbul, Turkey; 5Food and Agriculture Organization, King Abdulaziz Road, 558, Riyadh 11421, Saudi Arabia

**Keywords:** free phenolic, bound phenolic, purple wheat, gallic acid

## Abstract

The total phenolic content, phenolic compositions, and antioxidant capacity in the grain of 40 purple wheat genotypes were studied. In this study, purple wheats were investigated in terms of their composition of free and bound phenolic acids and 2,2-diphenyl-1-picrylhydrazyl radical scavenging capacity. The free phenolic content ranged from 164.25 to 271.05 mg GAE/100 g DW and the bound phenolic content was between 182.89–565.62 mg GAE/100 g wheat. The total phenolic content of purple wheat samples ranged from 352.65 to 771.83 mg GAE/100 g wheat. Gallic acid, protocatechuic acid, catechin, 4-hydroxybenzoic acid, syringic acid, ellagic acid, *m*-coumaric acid, *o*-coumaric acid, chrysin, caffeic acid, *p*-coumaric acid, ferulic acid, quercetin, kaempferol, rutin, sinapic acid, and chlorogenic acid were detected by HPLC system. Gallic acid, benzoic acid derivatives, and dominant phenolics, which are frequently found in cereals, were also dominant in purple wheat samples and were found in free fractions. The antioxidant capacity was assessed using the DPPH method. The antioxidant capacity (AA%) in the free phenolic extracts of the purple wheats was between 39.7% and 59.5%, and the AA% values of bound phenolic extract of the purple wheat varied between 42.6% and 62.7%. This study suggested that purple wheat samples have high phenolic compound content as antioxidant potential and therefore consumption of purple wheat-containing food products may provide health benefits.

## 1. Introduction

Wheat (*Triticum aestivum*) is a cereal crop that belongs to the Poaceae (*Gramineae*) family. It is one of the most important foods for humans [1]. Wheat grain is commonly processed into flour for its particular characteristics that are suitable for specific products such as bread, cake, pasta, noodle, and other bakery goods. The majority of the wheat on the market have white or red colored grain. Some unique wheat cultivars, such as purple and blue wheat grains, are also marketed commercially, but in small quantities [2,3]. Customers and grain producers are becoming more interested in colored cereal grains as a result of their potential health benefits.

Wheat contains phenolic compounds as secondary metabolites for typical plant functions. Phenolic acids and flavonoids are the major phenolic compounds found in wheat. There are two types of phenolic acids:hydroxybenzoic acid and hydroxycinnamic acid derivatives. Vanillic, syringic, *p*-hydroxybenzoic, and gallic acids are hydroxybenzoic acid derivatives while ferulic, *p*-coumaric, caffeic, and sinapic acids are the derivatives of hydroxycinnamic acid. The primary and most prevalent phenolic acid in wheat grain is ferulic acid. Vanillic, syringic, *o*-coumaric, *p*-coumaric, salicylic, and sinapic acids are also present in wheat in a smaller concentration [4]. The phenolic acids are found in three different forms: free, conjugated, and insoluble bound. The total concentration and composition of phenolic acids in wheat are affected by a number of parameters, including the type of wheat, growth conditions, morphology, and harvest time [5].

Recently, colored wheats rich in phytochemicals gain popularity around the world owing to the associated health benefits. For the first time, purple-grained varieties of wheat were released in New Zealand to improve color and texture of whole-grained bread [6]. In Europe the whole-grained bread products made from purple wheat have been produced since 2006 (The Kornspitz Company, Asten, Austria) under the brand “PurPur”. In Canada products from purple-grained wheat are sold under InfraReady Products brand (Saskatoon), of the registered trademark AnthoGrainTM (InfraReady Products Ltd., Saskatchewan, SK, Canada). In China, wheat grain with intense anthocyanin color was used for the production of noodles and soy sauce [7,8,9].

Colored wheat contains other phenolic compounds like phenolic acids which were also characterized by antioxidant capacity in non-pigmented wheat [10]. The contribution of bound phenolic acids to the total phenolic content and antioxidant capacity is significantly higher than that of free phenolic acids. Ferulic acid, a phenolic compound, is the main antioxidant of purple-grained wheat. White (refined) wheat flour, in which there is no bran, does not contain phenolic compounds [11,12]. Different amounts of phytochemicals such as anthocyanins, carotenoids, flavonoids, and some phenolic compounds are responsible for the color of wheat. Purple wheat has been a good candidate for developing functional food and enriching bakery products since the 19th century [13,14]. The purple grain color is caused by interaction of two complementary genes, Pp3 and Pp-1, mapped on a chromosome 2A and on 7B/7D chromosomes, respectively (7B in tetraploid T. durum (Pp-B1), and on the *T. aestivum* chromosome 7D (Pp-D1) [15,16]. These genes were transferred to common wheat from tetraploid wheat *Triticum turgidum* L. subsp. *abyssinicum Vavilov*, which originated from the Abyssinian region in Ethiopia [17,18].

Consumption of foods rich in phenolic acids can prevent the risk of some chronic health conditions such as type 2 diabetes, cardiovascular disease, and cancer. These health benefits can also be attributed in part to the phenolic acids found in wheat varieties [3]. Commercial production of value-added wheat varieties rich in health-beneficial components such as phenolic acids for making nutraceuticals and other functional foods are the concerns of recent studies [4]. Due to the health-promoting effects of phenolic acids and flavonoids, the increase of their content in mature kernels is of great interest and a potential target for wheat breeding programs. The production of wheat varieties with a high level of these bioactive phytochemicals is important for enhancing of the nutritional value of bread-making foods, expanding the local and export potential of the grain market [19,20].

The wheat samples used in this study are new varieties and advanced lines developed by classical plant breeding methods in Western Siberia. We have not encountered any studies on the phenolic composition of these or associated wheat samples in the related literature. After these investigations promising purple-grained lines characterized by high antioxidant capacity and ferulic acid content are received which are candidates for selection of commercial cultivars adapted to the conditions of the West Siberian region and suitable for functional food production.

The aim of this study was to identify the best accessions on the phenolic contents with high antioxidant capacity for their further evaluation in the nurseries of breeding process, and the search for a marker for antioxidants and/or phenolic content in wheat. The variation in the grain color is quite wide (purple, dark purple, light brown, red, blue and black) and further investigations on the material by the co-authors are expected to reveal marker (s) for these significant grain characteristics.

## 2. Materials and Methods

### 2.1. Plant Samples

A description of the plant materials is given in Table 1. The studied lines were obtained by a two-stage crossing of donor-colored lines with recurrent promising varieties of Siberian selection Element 22, Aina, Tobol’skaya, and breeding line BW49880 (CIMMYT, INT). The anthocyanin pericarp lines were selected using phenological markers of purple coloration of grains and dark-red coleoptiles, as well as using molecular SSR markers flanking the Pp-1D and Pp3 genes [21].

Near-isogenic lines, i:S29_Pp-D1Pp3P i:S29_Pp-D1Pp3PF (abbreviation—i:S29P and i:S29PF), inherited dark-red coleoptile color and purple pericarp color from their donor spring breeding lines Purple (k-46990, Australia) and Purple Feed (k-49426, Canada), respectively, in the genetic background of red-grained Russian spring bread wheat cv Saratovskaya 29 (S29) were created early by Arbuzova et al. (1998) [22]. These two lines with dominant alleles of genes Pp-1D and Pp3 on the chromosomes 7D and 2A were used as donors of the anthocyanin pericarp pigmentation.

The anthocyanin pigmentation in the aleurone layer of the grains was a substituted blue-grain wheat line based on the Saratovskaya 29 variety, carrying the Thinopyrum ponticum (Podp., syn. Agropyron elongatum Host., Elytrigia pontica Podp., Holub) chromosome 4Th instead of chromosome 4D (s:S29_4Th/4D), containing the dominant allele of the Ba gene (Blue aleuron) [23]. A two-stage crossing was carried out with recurrent promising varieties of Siberian breeding and breeding line BW 49880. Hybrid plants of the F1 generation had a mosaic light blue and white grain color in the ear. Blue grains were selected for further cultivation and pollination of recurrent parental forms in the F2 generation. Hybrid plants of the BC1F1 generation had mosaic light blue and white grains in the ear. Blue grains were selected for further cultivation. In the BC1F2 generation, blue grain plants were selected that were homozygous for the Ba gene and had homogeneous ears with well-made dark blue-colored grains. It should be noted that immature small grains may not have the blue color that appears during ripening.

In parallel, the selection of promising samples for the characteristics of high protein content, productivity and disease resistance was carried out on the basis of Omsk State Agrarian University [21]. To increase the anthocyanin content in wheat grains, we crossed plants with a blue-aleuron grained with purple-pericarp grained plants based on the Siberian cv Element 22 and breeding line BW49880. The parent varieties Saratovskaya 29, Element 22 had a red color of the seed coat, which is associated with the presence of an increased content of proanthocyanidins. Traditionally, the cultivation of red-grain wheat varieties is associated with increased climate humidity to prevent the germination of freshly ripened grains in the spikes. Thus, hybrids obtained by crossing blue-grained and purple-grained plants from red-grained varieties had black grains. The breeding line BW49880 had white grains without an increased content of proanthocyanidins in the grain coat. Hybrids obtained from crossing purple-grained and blue-grained plants from this line with the lines of red-grained cv Saratovskaya 29 split into 2 forms by color. Black hybrids (Black) contained proanthocyanidins in the seed coat, and dark purple hybrids (Dark purple) did not. Their seed coat between the pericarp and the aleurone layer is white. Light brown hybrid line BW 49880 does not seem to accumulate zinc.

Purple-grained awnless tetraploid hybrid emmer was obtained after a two-stage crossing of purple-grained T. durum Tri15744 (Gathersleben collection, Germany) with naked spelled. T. dicoccum cv Gremme (Tatarstan, Russia) and followed by a double crossing with the awnless sample of red-grained and red-spike emmer T. dicoccum k-25516 (Volga river selection, Chuvashia, Russia), resistant to stem and leaf rust with unknown genes.

#### 2.1.1. Field Trial Description

In 2021 at total 7 red-grained cultivars of spring bread wheat and 33 lines BC_1_F_3-8_ with colored grains were phenotyped in 1 m^2^ plots in the quality collection nursery (QCN) at the Omsk SAU (55°02′ N, 73°32′ E; Omsk, Russia). Field trials utilized a randomized complete block design with three replicates. Sowing, in the middle of May, was carried out with a SSFK-7 seeder with a sowing rate of 500 seeds per 1 m^2^. The predecessor was fallow. The cultivars Pamyati Azieva, Duet, and Element 22, as local checks, were sown every 20 numbers. The harvest was carried out in late August. Weather factors in 2021 led to not favorable conditions for moisture content in the soil: the average temperature was 16.6 °C and amount of precipitation—166 mm during the growing season, and therefore there was no damage of wheat crops with leaf and stem rust. The thousand kernel weight of the best lines with colored grains varied from 31.7 to 39.1 g. The grain yield was 422–597 g per 1 m^2^.

#### 2.1.2. Chemical Material

Hexane, acetone, diethyl ether, ethyl acetate, Folin–Ciocalteu reagent, 1,1-diphenyl-2-picryl-hydrazil (DPPH) were analytical quality and purchased from Sigma-Aldrich (Sigma-Aldrich Inc., Bornem, Belgium). Gallic acid was purchased from ICN Biomedicals, Inc. (ICN Biomedicals, Inc., Aurora, OH). Methanol (analytical grade), copper (II) chloride, absolute ethyl alcohol (EtOH), ammonium acetate (NH4Ac), and glacial acetic acid were purchased from Merck (Merck KGaA, Darmstadt, Germany).

### 2.2. Grain Samples and Sample Preparation

All grain (Table 1) was ground with a laboratory grinder (CemotecTM, CM290, Hillerod, Denmark) for two minutes.

### 2.3. Extraction

#### 2.3.1. Defatting Wheat Samples

Before starting the analysis, the oil is removed from the samples using hexane. Hexane is added to the samples at a ratio of 1:5 (*w*/*v*) and mixed with a vortex (5 mL hexane is added to 1 g sample). It is shaken at 200 rpm in a shaker (MK200D, Yamato Scientific Co., Ltd., Tokyo, Japan) for 10 min. It is centrifuged for 5 min at 2500× *g* (Centrifuge, Heraeus, Multifuge X3 FR, Thermo Scientific, Osterode am Harz, Germany). The remaining hexane is removed with a pipette and hexane is added again. This process is repeated 3 times. The defatted samples are dried on filter paper under a fume hood for 12 h.

#### 2.3.2. Extraction of Free Phenolic Compounds

A mixture of acetone-water (1:1) was used for the extraction of phenolic compounds of purple wheat samples. Firstly, wheat samples were ground and defatted. 0.5 g defatted wheat samples was added to 5 mL acetone—water mix to dissolve phenolic compounds. The extraction process was carried out for 1 h by using shaking water bath at a speed of 200 rpm and temperature of 25 °C. Then, the mixture was centrifuged at 2500× *g* (+4 °C) for 10 min (Centrifuge, Heraeus, Multifuge X3 FR, Thermo Scientific, Osterode am Harz, Germany). This process was repeated 3 times in total. Supernatants (liquid part) were taken with the help of a pipette and combined in a separate tube. The liquid portion was kept in the refrigerator at +4 °C, wrapped in aluminum foil, until the next stage. The solid part (pellet) was placed in the drying cabinet (30 °C) with its lid open and kept there overnight. Solvents were removed using a rotary evaporator (Hei-VAP Advantage, Heidolph, Schwabach, Germany). The water bath of the evaporator was adjusted to 40 °C. The rotation speed and pressure were 60 rpm and 65 mbar, respectively. The free phenolic compounds obtained were dissolved in methanol. 2 mL of methanol was added into the round evaporation flask. It was kept closed and wrapped in aluminum foil for 15 min. It was shaken at one-minute intervals to dissolve the phenolic compounds. The samples were taken into 4mL amber-colored vials and kept at −18 °C until analysis.

#### 2.3.3. Extraction of Bound Phenolics

The remaining solid (pellet) is alkaline hydrolyzed with 20 mL of 2 N NaOH for 4 h in a shaker at 200 rpm. The pH is adjusted to 1.8–2.2 using 6M HCl. To remove the remaining free fatty acids, hexane is applied 5 times. Each time 10 mL of hexane is added and mixed in a vortex and shaken at 200 rpm in a mixer for 10 min and centrifuged at 4000× g for 10 min (Centrifuge, Heraeus, Multifuge X3 FR, Thermo Scientific, Germany). The remaining hexane is removed with the help of a pipette. Extraction is then carried out with 10 mL of diethyl ether-ethyl acetate (1:1, *v*/*v*). It is mixed by vortex and shaken at 200 rpm for 15 min and centrifuged at 4000× *g* for 10 min. The remaining diethyl ether-ethyl acetate portion is separated with the help of a pipette. This process is repeated 6 times and all diethyl ether-ethyl acetate fractions are combined; the solvents are removed with a rotary evaporator (Hei-VAP Advantage, Heidolph, Germany). The resulting bound phenolic compounds are dissolved in methanol. 2 mL of methanol is added into the round evaporation flask. It is kept closed and wrapped in aluminum foil for 10–15 min. It is shaken at one-minute intervals to dissolve the phenolic compounds. The samples are taken into 4 mL amber-colored vials and kept at −18 °C until analysis.

### 2.4. Total Phenolic Contents (Free and Bound)

Free and bound phenolic compounds are determined and the total phenolic content is calculated by summing these values.

The free and bound phenolic contents were determined by the Folin-Ciocalteu method with some modification. 100 µL of extracts in methanol was mixed with 500 µL of Folin-Ciocalteu reagent (2 N) and 1.5 mL of Na_2_CO_3_ solution (200 g/L) and 7.9 mL distilled water. For the blank sample, 100 µL of methanol was added instead of extract. After 120 min standing in dark, it is centrifuged at 4000× *g* for 5 min. Then, the absorbance was measured at 760 nm by Shimadzu 150 UV-1800 spectrophotometer (Kyoto, Japan). The total phenolic contents were calculated based on the calibration curve of gallic acid and expressed as gallic acid equivalents (GAE), in ppm.

### 2.5. Measurement of Individual Phenolic Acids

The extracts were refrigerated overnight and recentrifuged and then filtered through a 0.22 µm filter. Chromatographic analyses were performed on an Agilent 1200 HPLC system consisting of a photodiode array detector, quaternary pump, auto sampler, and column oven. Phenolic acids (gallic acid, protocatechuic acid, catechin, 4-hydroxybenzoic acid, syringic acid, ellagic acid, *m*-coumaric acid, *o*-coumaric acid, chrysin, cafeic acid, *p*-coumaric acid, ferulic acid, myricetin, quercetin, kaempferol, rutin, sinapic acid, and chlorogenic acid) were separated on a Waters Atlantis C18 column (250 mm × 4.6 mm, 5 μm) using a linear gradient elution program with a mobile phase containing solvent A (acetic acid/H_2_O, 1:99, *v*/*v*) and solvent B (acetic acid/acetonitrile, 1:99, *v*/*v*) at a flow rate of 1 mL/min. The solvent gradient was programmed as follows: linear-gradient elution from 10% B to 60% B, 0–15 min; isocratic elution of 60% B, 15–20 min; linear gradient elution from 60% B to 10% B, 20–25 min; isocratic elution of 10% B, 25–30 min. The chromatograms were recorded at 278 nm, 320 nm, and 360 nm by monitoring spectra within the wavelength range 190–400 nm. Identification of phenolic acids was accomplished by comparing the retention time and absorption spectra of peaks in wheat samples to those of standard compounds. The quantitation of phenolic acids was based on calibration curves built for each of the compounds identified in the samples.

### 2.6. DPPH Radical Scavenging Capacity

Briefly, 100 μL of wheat extract was added to 3.9 mL of freshly prepared DPPH radical solution. After 60 min of reaction at room temperature, the absorbance of the solution was measured at 515 nm. DPPH radical scavenging capacity was expressed as antioxidant capacity (AA%).
Antioxidant Capacity (AA%) = [(A517 of control − A517 of sample)/A517 of control] × 100

### 2.7. Statistical Analysis

All experiments were carried out in triplicate, and the data were reported as the mean ± standard deviation. Statistical analysis was performed with SPSS Statistics Software (IBM version 20, Armonk, NY, USA). The one-way analysis of variance (ANOVA) was used to compare the contents of the bioactive compounds of the different wheat types. The significance of mean differences was determined by Duncan’s’ post hoc test and *t*-test at a 0.5 confidence interval.

## 3. Results and Discussion

### 3.1. Total Phenolic Content

The free, bound, and total phenolic content values of purple wheat varieties are given in Table 2. The amounts of free, bound and total phenolic compounds of wheat the samples varied between 164.03–271.05, 182.89–565.62 and 352.65–771.83 mg GAE/100g wheat, respectively. Yu and Beta [24] reported lower phenolic values, while Giordano, et al. [25] found higher results in the wheat samples they studied. Yu and Beta [24] reported that the free phenolic compounds of Indigo (97% purple wheat and 3% yellow wheat) wheat flour was 77.74 mg GAE/100g, its bound and total phenolic compounds were 91.87 mg GAE/100g and 169.61 mg GAE/100g. Also, Liu, et al. [3] found that purple wheat contains the highest TFC in the range of 21.59 to 102.95 mg of catechin equivalents per 100 g of wheat grain. The lower total phenolic content of the purple wheat varieties was reported from Okarter, et al. [26] study. The differences in total phenolic content reported in the literature can be explained by different wheat varieties and applied extraction techniques [13,24].

There was a significant difference between the total phenolic contents of wheat cultivars (*p* < 0.05). The samples w33 and w25 were the wheat cultivars containing the highest and lowest total phenolic content, respectively. The amount of bound phenolic was found to be higher than the amount of free phenolic in all wheat varieties. The bound phenolic compounds have important functions such as high antioxidant and antimicrobial properties in the colon [27]. The contribution of bound phenolics to total phenolics varied between 51% and 73%. In general, this ratio of bound phenolics is in agreement with previously published studies. Okarter, et al. [26] reported the ratio of bound phenolics to total phenolic content as 53–69%. In the study of Liyana-Pathirana and Shahidi [27], the bound phenolic contribution varies between 55–58%. In our study, w33 and w16 wheat samples were containing the highest bound phenolics.

### 3.2. Phenolic Acids Profile

Phenolic acids have many beneficial effects on human health due to their high antioxidant properties. On regular eating, phenolic acids also promote the anti-inflammation capacity of human beings. Therefore, the production of wheat varieties rich in phenolic acid has always been an important issue. Phenolic acids were also investigated in bread, durum and colored wheats by Dinelli, et al. [28], Liu, et al. [3], Nicoletti, et al. [29], and Sharma, et al. [30]. They found that phenolic acids are one of the most important phenolic groups. The profile of the phenolics in free and bound phenolic extracts of 40 different purple wheat samples and their concentrations are demonstrated in Table 3 and Table 4, respectively.

As can be seen from the table, the distributions of phenolic acids varied according to wheat varieties and whether phenolic acids were bound or free. Gallic acid, benzoic acid derivatives and predominant phenolics in cereals, was the predominant phenolic acid found in purple wheat samples and was found in the free fractions (Table 3). Free gallic acid content ranged from 15.41 to 95.28 μg/g of dry weight. The free phenolic fractions of w19 and w31 samples had the highest and lowest gallic acid content, respectively. The second most abundant phenolic acid found in free fraction is protocatechuic acid (6.93–66.35 μg/g of dry weight). Ellagic acid was found in free fractions of purple wheat samples (8.20–19.64 μg/g of dry weight) except for w27 and w30 samples. Catechin was found in high amounts in w9, w23 and w24 samples (25.79, 27.11, and 23.70 μg/g of dry weight, respectively).

Table 4 revealed that protocatechuic, ferulic, and gallic acid were the predominant phenolic acids found in the bound fractions of purple wheat samples. The amount of bound protocatechuic acid was 2.56–160.79 μg/g of dry weight. In bound fractions, some samples (w11, w18, w20 and w32) had high ferulic acid content (307.18, 423.92, 256.61, and 581.96 μg/g of dry weight, respectively). Ferulic acid, a hydroxycinnamic acid derivative, is the predominant phenolic acid found in whole grain and wheat samples [31,32,33]. Okarter, et al. [26] reported that the bound fraction contributed 87–97% to the total ferulic acid content. Liu, et al. [3] investigated the phenolic acid content of red, yellow, purple and common wheat and purple colored wheat contained the highest content of ferulic acid (81.38 mg/100 g). It was reported from previously mentioned published studies that the ferrulic acid is the major phenolic content in especially bound fraction. The reason for the difference in our study may be due to the extraction method, climatic conditions and breeding process applied in our study. It should also be taken into account that our study has only one year data. This can be seen as a weakness for our study. However, in previously punlished studies, phenolic characterization of different known colored wheat cultivars was performed. In our study, unlike these studies, the effect of the breeding process on the amount and distribution of phenolic compounds in the new purple-grained lines was investigated. The results of our study suggested that new lines can adapt to the Siberian region with different phenolic content could be obtained by breeding process and could be considered as a new commercial cultivars.

The results of this study showed that the wheat cultivars used in this study had high bound phenolic acid content and the bound phenolic acid distribution was different from free phenolic acids. Phenolic content varies among different wheat cultivars or accessions. The inter-genotypic variability of total phenolic content is 18.5% in wheat cultivars [34]. Paznocht, et al. [35] reported that the total phenolic acid content of 4 different colored wheat groups varied: blue aleurone > purple pericarp > yellow endosperm > red color (798 > 702 > 693 > 599 µg/g). Biofortified colored wheat (black, blue, and purple) had higher soluble and insoluble phenolic compounds, anthocyanin content, and antioxidant capacity in the order of black > blue > purple > white [36]. Ferulic acid prevailed in red and yellow kinds of wheat, vanillic in blue, and *p*-coumaric in purple kinds of wheat. The research conducted by Zhang, et al. [37] revealed that the blue wheat outer bran had the highest total bound phenolic acid of 3458.71 μg/g while the purple wheat shorts had the lowest of 1730.71 μg/g. Ma, et al. [38] reported that among phenolic acid compounds, bound ferulic acid, vanillic, and caffeic acid levels were significantly higher in purple wheat than in white and red wheat, while total soluble phenolic acid, soluble ferulic acid, and vanillic acid levels were significantly higher in purple and red wheat than in white wheat. Nine phenolic acid biosynthesis pathway genes (TaPAL1, TaPAL2, TaC3H1, TaC3H2, TaC4H, Ta4CL1, Ta4CL2, TaCOMT1, and TaCOMT2) exhibited three distinct expression patterns during grain filling, which may be related to the different phenolic acid levels. Also, some identified differentially expressed genes, such as flavonol synthase (TraesCS1A02G221200 and TraesCS1A02G320000), and glycosyltransferase genes (TraesCS7D02G198300 and TraesCS1D02G301700) may play an important role in phenolics biosynthesis [39].

### 3.3. Antioxidant Capacities with DPPH

The antioxidant capacity (AA%) by DPPH values of the purple wheat samples are presented in Table 5. While the antioxidant capacity (AA%) in the free phenolic extracts of the purple wheats was between 39.7% and 59.5%, the AA% values of bound phenolic extracts of the purple wheats varied between 42.6% and 62.7%. The w33 sample showed the highest AA% value for bound phenolic extract (62.7%) and the highest phenolic content of bound extract (565.62 mg GAE/100g). This result showed that a positive correlation was found between DPPH radical scavenging activity and the total phenolic content of purple wheat samples. On the other hand, w5 had the highest free antioxidant capacity, explaining that the free and bound phenolic extracts of purple wheat samples were found to exhibit different reaction kinetics compared with the antioxidant standard. There was a significant difference between the AA% of wheat cultivars (*p* < 0.05). These results support that purple wheat genotype and color affect antioxidant capacity, and these factors should be taken into consideration in the development of high-antioxidant wheat samples.

Colored wheats have a high antioxidant capacity against free oxygen radicals and DPPH radicals. DPPH radical scavenging capacity has been reported for other wheat samples, dark blue (*Triticum aestivum* L. cv. Hedong Wumai) and Swiss red wheat grain [40,41]. Hu, et al. [40] reported that the radical scavenging activity was 0.28 mmol Trolox equivalent/g dark blue grained wheat. When the results of this study were examined, it was seen that bound phenolic compounds were more effective than free phenolic ones in terms of DPPH radical scavenging activity. This indicated that bound phenolics of all fractions had higher antioxidant capacity than free phenolics. These results were derived from higher total phenolic contents in bound phenolics than in free phenolics [37].

## 4. Conclusions

In conclusion, free and bound extracts of 40 different purple wheat samples from different wheat genotypes differed significantly in their radical scavenging capacities against DPPH and in TPC, indicating the potential effect of genotype on the antioxidant properties of wheat samples. Also, each of purple wheat sample had a different type and amount of phenolic acids. In this study, ferulic acid may serve as a marker for quality control of wheat antioxidants or may be used to monitor wheat antioxidant processing. In bound fraction, w33 (line Element 22-Purple ^PF^ (2-2-7), BC_1_F_5_) had the highest antioxidant capacity and ferulic acid content. The results of this study indicated that the breeding process may bring about significant variation in the phenolic contents of wheats. The perspectives of this study demonstrate its successful application for development of purple-grained varieties of spring bread wheat, adapted to conditions of the West Siberian region, having high antioxidant capacity, and finally, suitable for functional food production. Further investigations on the material are expected to reveal marker (s) for these important grain characteristics.

## Figures and Tables

**Table 1 foods-11-02515-t001:** Purple wheat sample types used in this study.

#	Cultivar/Line, Grain Color and Generation	Origin
w1	*cv* Pam’yati Azieva-Red, St	Saratovskaya29/Lutescens 99/80-1.7
w2	*cv* Duet-Red, St	Individual selection from the hybrid population Erythrospermum 59//Tselinnaya 20/ANK 102
w3	*cv* Element 22-Red, St	(Granıt X Saratovskaya29) X [Erythrospermum59 X (Tselınnaya 20 X Tertsıya)]
w4	*cv* Tobol’skaya-Red, St Red	Lutescens 123-S/Omskaya20
w5	*cv* Saratovskaya 29-Red, St	Albıdum 24/Lutescens 55/11
w6	*cv* Aina-Red, St	Tselınnaya Yubılıenaya/2*Pastor/3/Babax/Lr43//Babax
w7	*cv* Seri 82-Red, St	Kavkaz/(Sıb) Buho//Kalyansona/Bluebırd
w8	line10 Element_22-Blue^4Th^ (1)	S29_(4Th/4D)/Element_22
w9	line 11 Element_22 Blue^4Th^ (1)	S29_(4Th/4D)/Element_22
w10	*cv* Balda *T. dicoccum*	Individual selection from the hybrid population Belka (*T. dicoccum*)/Svetlana (*T. durum*)//Belka
w11	line Emmer-Purple^Tri15744^	*T. durum* Tri15744/*T. dicoccum* *cv* Gremme//*2 *T. dicoccum* K-25516
w12	*cv* Saratovskaya 29-Purple^PF^ (5)	i:S29_PF
w13	line Aina–Purple^PF^, BC_1_F_8_	Aina *2/i:S29 PF
w14	line Element 22-Purple^PF^ (8), BC_1_F_8_	Element 22 *2/i:S29_PF
w15	line Saratovskaya 29-Purple^PF^ (2)	i:S29_PF
w16	line Tobol’skaya–Purple^PF^, BC_1_F_8_	Tobol’skaya *2/i:S29_PF
w17	line BW 49880–Dark purple^P^, F_4_	BW 49880 *2/S29_4Th)//BW 49880 *2/i:S29_P
w18	line Element 22-Black, F_4_	Element 22 *2/S29_4Th)//Element 22 *2/i:S29_PF
w19	line BW 49880–Purple^PF^, BC_1_F_8_	BW 49880 *2/i:S29_PF
w20	line BW 49880–Black^P+4Th^, F_4_	BW 49880 *2/S29_4Th)//BW 49880 *2/i:S29_P
w21	line BW 49880-Dark Purple^P+4Th^, F_4_	BW 49880 *2/S29_4Th)//BW 49880 *2/i:S29_P
w22	line Element 22-Purple^PF^, BC_1_F_8_	Element 22 *2/i:S29_PF
w23	line BW 49880-Dark Purple^P+4Th^, F_4_	BW 49880 *2/S29_4Th)//BW 49880 *2/i:S29_P
w24	line BW 49880-Light brown^P+4Th^, F_4_	BW 49880 *2/S29_4Th)//BW 49880 *2/i:S29_P
w25	line BW 49880-Purple^PF^ (10-7), BC_1_F_4_	BW 49880 *2/i:S29 PF
w26	line BW 49880-Purple^PF^ (10-4-1), BC_1_F_5_	BW 49880 *2/i:S29 PF
w27	line Element 22-Purple^PF^, BC_1_F_3_	Element 22 *2/i:S29_PF
w28	line Element 22-Purple^PF^, BC_1_F_3_	Element 22 *2/i:S29_PF
w29	line Element 22-Purple^PF^ (2-7), BC_1_F_4_	Element 22 *2/i:S29_PF
w30	line Element 22-Purple^PF^ (2-8), BC_1_F_4_-	Element 22 *2/i:S29_PF
w31	line Element 22-Purple^PF^ (2-10), BC_1_F_4_	Element 22 *2/i:S29_PF
w32	line Element 22-Purple^PF^ (2-2-1), BC_1_F_5_	Element 22 *2/i:S29_PF
w33	line Element 22-Purple^PF^ (2-2-7), BC_1_F_5_	Element 22 *2/i:S29_PF
w34	line Element 22-Purple^PF^ (2-3-8), BC_1_F_5_	Element 22 *2/i:S29_PF
w35	line Element 22-Purple^PF^ (2-3-12), BC_1_F_5_	Element 22 *2/i:S29_PF
w36	line Element 22-Purple^PF^, (2-3-17) BC_1_F_5_	Element 22 *2/i:S29_PF
w37	line Element 22-Purple (2-4-6), BC_1_F_5_	Element 22 *2/i:S29_PF
w38	line Element 22-Purple^PF^, (2-5-1) BC_1_F_5_	Element 22 *2/i:S29_PF
w39	line Element 22-Purple^PF^, (2-5-4), BC_1_F_5_	Element 22 *2/i:S29_PF
w40	line Element 22-Purple^PF^, (2-5-14), BC_1_F_5_	Element 22 *2/i:S29_PF

**Table 2 foods-11-02515-t002:** Free, bound, and total phenolic content of purple wheat samples (mg GAE/100g).

# of Sample	Free Phenolics	Bound Phenolics	Total Phenolics
w1	188.62 ± 0.5 ^JKL^	399.18 ± 0.1 ^F^	587.80 ± 0.6 ^F^
w2	221.04 ± 0.1 ^C^	487.21 ± 0.6 ^C^	708.25 ± 0.7 ^B^
w3	243.97 ± 0.6 ^B^	386.55 ± 1.0 ^G^	630.52 ± 1.6 ^D^
w4	203.66 ± 0.7 ^EF^	226.76 ± 0.1 ^U^	430.42 ± 0.8 ^U^
w5	271.05 ± 0.0 ^A^	307.29 ± 0.2 ^K^	578.34 ± 0.2 ^G^
w6	200.91 ± 0.2 ^EFG^	249.65 ± 0.1 ^R^	450.56 ± 0.3 ^QRS^
w7	211.29 ± 0.3 ^D^	234.39 ± 0.7 ^T^	445.68 ± 0.0 ^ST^
w8	246.89 ± 0.1 ^B^	519.85 ± 0.8 ^B^	766.74 ± 0.9 ^A^
w9	204.38 ± 0.5 ^EF^	252.62 ± 0.4 ^R^	457.00 ± 0.9 ^Q^
w10	185.23 ± 0.9 ^LMN^	210.23 ± 0.5 ^W^	395.46 ± 1.4 ^W^
w11	195.19 ± 0.6 ^GHI^	283.13 ± 0.8 ^P^	478.32 ± 1.4 ^NOP^
w12	180.78 ± 0.8 ^NOP^	258.97 ± 0.3 ^Q^	439.75 ± 1.1 ^T^
w13	221.57 ± 0.2 ^C^	235.45 ± 0.1 ^T^	457.13 ± 0.3 ^Q^
w14	182.68 ± 0.0 ^MNO^	192.89 ± 0.1 ^Z^	375.58 ± 0.1 ^Y^
w15	203.66 ± 0.1 ^EF^	212.75 ± 0.0 ^W^	416.41 ± 0.1 ^V^
w16	184.80 ± 0.1 ^LMN^	192.85 ± 0.1 ^X^	377.66 ± 0.2 ^X^
w17	211.08 ± 0.0 ^D^	243.50 ± 0.4 ^S^	454.58 ± 0.4 ^QR^
w18	197.31 ± 0.0 ^GH^	322.97 ± 0.1 ^I^	520.28 ± 0.1 ^I^
w19	194.34 ± 0.0 ^HIJ^	232.27 ± 0.0 ^T^	426.61 ± 0.0 ^U^
w20	217.65 ± 0.1 ^C^	258.80 ± 0.1 ^Q^	476.45 ± 0.2 ^OP^
w21	190.31 ± 0.2 ^IJKL^	285.68 ± 0.2 ^OP^	475.99 ± 0.4 ^OP^
w22	168.70 ± 0.0 ^ST^	190.31 ± 0.4 ^XY^	359.01 ± 0.4 ^Y^
w23	191.16 ± 0.2 ^IJK^	287.37 ± 0.0 ^NO^	478.53 ± 0.2 ^NO^
w24	194.97 ± 0.0 ^HI^	309.20 ± 0.1 ^K^	504.17 ± 0.1 ^K^
w25	164.03 ± 0.2 ^T^	188.62 ± 0.0 ^Y^	352.65 ± 0.2 ^Z^
w26	175.48 ± 0.1 ^PQR^	309.20 ± 0.0 ^K^	484.68 ± 0.1 ^MN^
w27	176.11 ± 0.2 ^PQ^	219.56 ± 0.3 ^V^	395.67 ± 0.5 ^W^
w28	184.80 ± 0.2 ^LMN^	192.62 ± 0.0 ^Y^	374.42 ± 0.2 ^X^
w29	187.98 ± 0.0 ^KLM^	300.93 ± 0.2 ^L^	488.91 ± 0.2 ^LM^
w30	247.70 ± 0.3 ^B^	431.26 ± 0.1 ^D^	678.96 ± 0.4 ^C^
w31	177.60 ± 0.1 ^OP^	338.66 ± 0.2 ^H^	516.25 ± 0.3 ^IJ^
w32	170.82 ± 0.1 ^QRS^	341.62 ± 0.7 ^H^	512.44 ± 0.8 ^J^
w33	206.21 ± 0.0 ^DE^	565.62 ± 0.3 ^A^	771.83 ± 0.3 ^A^
w34	171.66 ± 0.0 ^QRS^	427.24 ± 0.6 ^E^	598.90 ± 0.6 ^E^
w35	167.21 ± 0.3 ^RST^	192.85 ± 0.2 ^X^	360.07 ± 0.5 ^Y^
w36	182.05 ± 0.0 ^NO^	289.49 ± 0.4 ^N^	471.54 ± 0.4 ^P^
w37	211.50 ± 0.2 ^D^	317.89 ± 0.5 ^J^	529.39 ± 0.7 ^H^
w38	199.42 ± 0.1 ^FGH^	249.60 ± 0.0 ^R^	449.07 ± 0.1 ^RS^
w39	194.55 ± 0.0 ^HI^	297.33 ± 0.2 ^M^	491.88 ± 0.2 ^L^
w40	164.25 ± 0.0 ^T^	189.68 ± 0.1 ^XY^	353.92 ± 0.1 ^Z^

Values with different letters in each wheat sample and fraction indicate statistically significant differences at *p* < 0.05.

**Table 3 foods-11-02515-t003:** Free phenolic compounds of purple wheat samples (μg/g of dry weight).

Sample	Phenolic Compounds													
	Gallic Acid	Protocate-Chuic Acid	Catechin	4-Hydroxy-Benzoic Acid	Syringic Acid	Ellagic Acid	*m*-Coumaric Acid	*o*-Coumaric Acid	Cafeic Acid	*p*-Coumaric Acid	Ferulic Acid	Rutin	Sinapic Acid	Chlorogenic Acid
w1	74.35	13.55	1.86	nd	2.20	9.65	nd	nd	4.54	1.08	6.89	5.86	4.49	0.96
w2	21.14	22.03	2.08	1.51	2.47	11.60	nd	nd	4.92	1.35	9.18	8.30	6.04	1.12
w3	35.10	54.27	2.22	nd	0.42	8.85	0.62	nd	4.64	1.12	6.58	5.53	4.70	1.03
w4	62.82	28.01	3.67	1.11	1.78	12.03	nd	nd	6.42	2.23	9.83	9.00	11.09	2.36
w5	88.45	31.94	4.56	nd	2.74	10.96	nd	nd	4.25	1.01	8.21	7.27	4.03	1.29
w6	50.09	37.17	3.89	0.48	2.15	13.93	nd	nd	4.59	0.93	10.51	9.72	3.58	0.94
w7	26.54	38.45	nd	0.97	1.99	10.33	nd	nd	4.43	1.69	8.65	7.74	7.98	1.16
w8	35.20	19.04	nd	0.60	1.40	8.20	nd	nd	4.47	0.99	6.91	5.89	3.95	1.21
w9	34.26	66.35	25.79	0.52	nd	16.53	nd	nd	6.08	1.77	13.54	12.96	8.48	2.09
w10	74.25	17.62	8.27	nd	5.76	14.26	nd	nd	4.53	0.87	6.83	5.80	3.26	1.05
w11	91.52	29.15	nd	0.64	1.84	11.24	nd	nd	4.20	1.13	7.62	6.64	4.76	1.19
w12	66.42	16.45	nd	nd	0.28	8.48	nd	nd	4.52	1.28	5.98	4.90	5.65	0.92
w13	50.25	21.88	nd	nd	nd	17.01	nd	nd	4.98	1.41	5.61	4.50	6.35	3.95
w14	29.36	17.20	2.16	0.76	1.79	10.58	1.87	nd	5.20	0.85	7.80	6.84	3.14	1.41
w15	10.12	45.40	nd	0.37	0.44	11.06	1.60	nd	5.53	2.09	8.98	8.10	10.28	1.53
w16	39.99	39.92	nd	nd	1.44	11.05	1.60	nd	5.78	1.52	7.97	7.02	7.03	2.16
w17	86.40	50.86	nd	3.95	nd	13.18	1.27	nd	6.59	4.00	13.05	12.43	21.34	3.01
w18	39.97	11.85	nd	nd	nd	7.76	2.03	nd	5.90	3.12	9.23	8.36	16.23	2.75
w19	15.41	35.08	3.24	1.61	nd	11.54	1.46	nd	6.64	5.29	10.28	9.48	28.80	6.96
w20	26.35	35.46	nd	7.74	0.48	15.25	1.53	nd	6.13	2.79	10.64	9.86	14.38	5.49
w21	41.25	44.87	nd	4.50	2.03	11.59	1.46	nd	6.66	3.77	10.87	10.10	20.01	7.65
w22	54.34	10.08	nd	nd	nd	14.64	1.92	nd	4.39	1.31	4.43	3.24	5.77	0.82
w23	34.94	20.47	27.11	nd	0.38	8.28	1.79	nd	4.55	3.14	5.88	4.78	16.40	3.48
w24	65.10	18.09	23.70	nd	0.24	8.69	1.80	nd	4.96	4.99	8.76	7.86	27.06	6.01
w25	68.46	6.93	1.74	nd	nd	16.98	1.88	nd	4.52	1.47	5.24	4.11	6.75	0.85
w26	41.27	10.93	2.60	nd	0.11	11.69	2.04	nd	4.21	2.05	6.00	4.91	10.06	1.85
w27	29.73	13.13	nd	nd	0.93	nd	2.05	nd	4.36	1.03	4.55	3.37	4.16	1.76
w28	21.53	34.04	3.43	nd	4.23	18.10	1.61	0.25	4.70	4.09	14.29	13.76	21.86	5.19
w29	52.97	18.99	3.89	nd	0.67	10.26	1.79	nd	4.45	1.95	7.96	7.00	9.50	3.05
w30	34.72	14.93	nd	nd	0.11	nd	1.91	nd	4.29	0.92	4.15	2.94	3.52	1.61
w31	95.28	10.07	nd	nd	0.62	17.48	1.85	nd	4.61	1.60	5.58	4.47	7.45	0.81
w32	46.65	27.25	nd	nd	0.14	10.21	1.94	nd	4.33	1.66	5.67	4.56	7.84	1.88
w33	53.38	12.12	nd	nd	0.1	12.33	1.69	1.12	4.37	2.1	6.71	5.67	10.34	1.28
w34	52.84	10.45	nd	nd	0.25	10.35	1.86	1.05	4.43	0.99	5.29	3.94	4.16	1.5
w35	36.34	22.16	nd	nd	0.31	14.72	nd	nd	4.29	1.52	4.01	7.03	2.8	1.86
w36	52.91	15.67	nd	nd	0.06	19.64	1.86	0.68	4.51	2.01	6.21	9.86	5.14	1.2
w37	42.38	15.45	1.51	nd	0.04	19.26	2.02	nd	4.37	1.8	5.72	8.6	4.62	1.23
w38	46.68	16.34	3.45	nd	1.39	10.16	nd	nd	4.47	2.33	6.56	11.7	4.62	1.2
w39	62.20	16.35	nd	nd	0.21	17.65	1.94	nd	4.33	2.01	5.68	9.83	4.58	1.19
w40	50.95	23.84	nd	nd	0.21	11.78	1.95	0.28	4.46	2.07	6.75	10.2	5.72	1.56

nd: not detected.

**Table 4 foods-11-02515-t004:** Bound phenolic compounds of purple wheat samples (μg/g of dry weight).

Sample	Phenolic Compounds													
	Gallic Acid	Protocate-Chuic Acid	Catechin	4-Hydroxy-Benzoic Acid	Syringic Acid	Ellagic Acid	*m*-Coumaric Acid	*o*-Coumaric Acid	Cafeic Acid	*p-Coumaric* Acid	Ferulic Acid	Rutin	Sinapic Acid	Chloro-Genic Acid
w1	11.74	nd	nd	nd	nd	nd	0.15	nd	nd	0.71	6.56	2.30	5.51	nd
w2	13.29	160.79	nd	nd	nd	14.39	nd	nd	nd	0.32	2.12	0.10	0.78	nd
w3	10.41	nd	nd	nd	nd	9.38	1.19	nd	4.12	1.15	19.04	4.89	18.82	nd
w4	11.40	49.41	nd	nd	nd	18.96	nd	nd	nd	0.54	4.60	1.33	3.43	nd
w5	11.67	15.99	nd	nd	nd	nd	nd	nd	nd	0.65	1.75	1.98	0.38	nd
w6	23.44	125.55	nd	nd	nd	nd	1.52	nd	nd	0.52	1.57	1.20	0.19	nd
w7	16.30	2.56	nd	nd	nd	8.25	0.42	nd	nd	0.92	5.32	3.53	4.19	nd
w8	nd	nd	nd	nd	nd	nd	0.81	nd	nd	0.92	4.13	3.55	2.92	nd
w9	11.61	73.02	nd	nd	nd	nd	1.03	nd	nd	0.47	3.61	0.94	2.37	nd
w10	nd	nd	nd	nd	nd	15.10	nd	nd	nd	0.49	3.43	1.06	2.17	nd
w11	11.72	10.96	nd	1.30	0.86	14.73	16.93	nd	5.26	9.56	307.18	53.47	325.96	0.96
w12	8.84	nd	nd	nd	nd	20.19	nd	nd	nd	0.43	5.67	0.71	4.56	nd
w13	14.36	63.35	nd	nd	nd	12.89	nd	nd	nd	0.43	2.99	0.69	1.70	nd
w14	11.40	24.62	nd	nd	nd	14.79	nd	nd	nd	0.47	3.15	0.66	1.87	nd
w15	11.77	90.18	nd	nd	nd	67.23	nd	nd	nd	1.34	23.45	5.95	23.52	nd
w16	7.75	32.48	nd	nd	nd	10.09	nd	nd	nd	0.37	1.81	0.35	0.45	nd
w17	nd	nd	nd	nd	nd	10.51	nd	nd	nd	0.35	1.94	0.25	0.59	nd
w18	11.92	5.60	nd	1.48	1.14	18.00	22.36	nd	5.31	12.49	423.92	70.42	450.41	1.15
w19	12.05	nd	nd	nd	nd	31.35	nd	nd	nd	0.65	8.60	1.99	7.68	0.82
w20	11.63	6.30	nd	1.23	0.87	7.94	16.77	nd	4.26	10.08	256.61	56.51	272.05	nd
w21	10.53	nd	nd	nd	nd	nd	nd	nd	nd	0.36	1.54	0.33	0.16	nd
w22	15.16	nd	nd	nd	nd	38.01	nd	nd	nd	0.74	11.10	2.51	10.35	nd
w23	10.70	16.08	nd	nd	nd	16.30	nd	nd	nd	0.58	4.46	1.58	3.27	0.85
w24	11.11	nd	nd	nd	nd	33.30	nd	nd	nd	0.91	10.13	3.51	9.32	0.88
w25	9.02	10.30	nd	nd	nd	12.96	nd	nd	nd	0.69	2.96	2.23	1.67	nd
w26	29.18	117.43	nd	nd	nd	43.45	nd	nd	nd	0.77	11.58	2.67	10.87	nd
w27	19.76	147.40	nd	nd	nd	27.08	nd	nd	nd	0.63	7.52	1.84	6.53	0.92
w28	9.75	nd	nd	nd	nd	63.55	nd	nd	nd	1.27	22.26	5.57	22.24	0.85
w29	11.07	nd	nd	nd	nd	15.18	nd	nd	nd	0.92	2.36	0.53	1.04	d
w30	nd	nd	nd	nd	nd	26.02	nd	nd	nd	1.10	2.89	0.73	1.60	nd
w31	9.48	55.18	nd	nd	nd	145.48	0.59	nd	nd	2.08	42.91	10.27	44.26	0.84
w32	10.81	7.71	nd	nd	2.98	31.88	40.20	nd	5.58	23.92	581.96	136.48	618.87	1.31
w33	15.18	nd	nd	nd	nd	46.06	nd	nd	4.11	1.18	15.46	5.04	15.00	nd
w34	11.07	nd	nd	nd	nd	9.57	16.82	nd	4.32	11.46	40.42	11.68	277.29	0.99
w35	16.88	49.62	nd	nd	nd	nd	nd	nd	nd	0.30	1.60	0.04	0.23	nd
w36	8.78	nd	nd	nd	nd	44.14	1.52	nd	nd	0.94	11.59	3.65	10.88	0.87
w37	9.79	103.07	nd	nd	nd	88.51	nd	nd	nd	1.52	4.99	1.39	25.57	nd
w38	20.04	156.82	nd	nd	nd	30.52	nd	nd	nd	0.73	8.14	2.43	7.19	nd
w39	12.03	29.12	nd	nd	nd	9.28	nd	nd	nd	0.77	1.71	2.66	0.35	nd
w40	nd	nd	nd	nd	nd	89.27	nd	nd	nd	1.74	22.79	8.29	22.81	nd

nd: not detected.

**Table 5 foods-11-02515-t005:** Antioxidant capacity (DPPH: radical scavenging capacity) of purple wheat samples.

# of Sample	Free (AA%)	Bound (AA%)
w1	47.9 ± 0.1 ^N^	59.6 ± 0.1 ^F^
w2	55.4 ± 0.2 ^E^	61.9 ± 0.2 ^C^
w3	57.8 ± 0.3 ^B^	59.6 ± 0.2 ^F^
w4	52.1 ± 0.0 ^J^	47.7 ± 0.3 ^U^
w5	59.5 ± 0.4 ^A^	55.3 ± 0.3 ^L^
w6	52.0 ± 0.3 ^J^	50.3 ± 0.3 ^R^
w7	54.1 ± 0.1 ^G^	48.8 ± 0.1 ^S^
w8	57.8 ± 0.4 ^B^	62.2 ± 0.0 ^B^
w9	53.6 ± 0.2 ^HI^	51.5 ± 0.1 ^Q^
w10	47.5 ± 0.2 ^O^	47.2 ± 0.2 ^V^
w11	50.8 ± 0.2 ^L^	54.5 ± 0.1 ^N^
w12	45.4 ± 0.2 ^R^	53.2 ± 0.0 ^P^
w13	55.8 ± 0.2 ^D^	48.9 ± 0.1 ^S^
w14	45.6 ± 0.3 ^QR^	42.6 ± 0.1 ^Z^
w15	52.1 ± 0.0 ^J^	46.9 ± 0.2 ^W^
w16	45.8 ± 0.0 ^PQ^	45.6 ± 0.0 ^X^
w17	53.9 ± 0.2 ^GH^	49.1 ± 0.0 ^S^
w18	51.4 ± 0.6 ^K^	57.6 ± 0.1 ^I^
w19	49.9 ± 0.1 ^M^	48.5 ± 0.1 ^T^
w20	54.4 ± 0.1 ^F^	50.2 ± 0.0 ^R^
w21	48.0 ± 0.0 ^N^	54.6 ± 0.3 ^N^
w22	42.7 ± 0.3 ^T^	45.4 ± 0.1 ^X^
w23	49.9 ± 0.3 ^M^	54.6 ± 0.2 ^N^
w24	50.7 ± 0.1 ^L^	56.1 ± 0.2 ^K^
w25	39.7 ± 0.1 ^V^	43.6 ± 0.1 ^Z^
w26	44.2 ± 0.2 ^S^	55.6 ± 0.0 ^L^
w27	44.2 ± 0.1 ^S^	47.7 ± 0.2 ^U^
w28	45.9 ± 0.2 ^P^	45.0 ± 0.2 ^Y^
w29	47.8 ± 0.0 ^N^	54.9 ± 0.4 ^M^
w30	56.7 ± 0.1 ^C^	61.2 ± 0.3 ^D^
w31	45.4 ± 0.4 ^R^	58.6 ± 0.2 ^H^
w32	44.0 ± 0.1 ^S^	58.9 ± 0.0 ^G^
w33	53.5 ± 0.0 ^I^	62.7 ± 0.2 ^A^
w34	44.0 ± 0.2 ^S^	60.3 ± 0.2 ^E^
w35	42.4 ± 0.0 ^T^	46.8 ± 0.0 ^W^
w36	45.4 ± 0.0 ^R^	54.6 ± 0.3 ^N^
w37	54.4 ± 0.0 ^F^	57.1 ± 0.2 ^J^
w38	51.5 ± 0.2 ^K^	53.9 ± 0.3 ^O^
w39	50.0 ± 0.2 ^M^	54.6 ± 0.0 ^N^
w40	41.7 ± 0.1 ^U^	45.4 ± 0.0 ^X^

Values with different letters in each wheat sample and fraction indicate statistically significant differences at *p* < 0.05.

## Data Availability

The data presented in this study are available on request from the corresponding author.

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
