# Peer review of "Antioxidant Capacity and Profiles of Phenolic Acids in Various Genotypes of Purple Wheat"

_foods, 2022, doi:10.3390/foods11162515_

Round 1

Reviewer 1 Report

The manuscript is generally well-written and touches an interesting subject of antioxidant activity and phenolic compounds’ profiles and contents in different genotypes of purple wheat. However, there are some aspects that need to be further explained and/or corrected.

 The manuscript title should be revised/corrected (incorrect: “Total Phenolic Compound”; “Phenolic Acid Profile of Phenolic Compounds”). Suggestion: Antioxidant Activity and Profiles of Phenolic Compounds in Various Genotypes of Purple Wheat.

In the Introduction section it is mentioned (and repeated a couple of times) that the primary and most prevalent phenolic acid in wheat grain (also in purple wheat) is ferulic acid, while the presented study indicates other acids as predominant in the tested samples (gallic acid and protocatechuic acid in free and bound fractions of wheat samples, respectively). This should be discussed and explained.

According to the methods description, field replications were included in the study design, but another important factor – season (=weather conditions) – was not taken into account. The study was carried out for one year only. It is well known that season-to-season differences in the weather conditions can strongly impact crop composition, including the phenolic contents and profiles, thus the study should be carried out for at least two or three seasons to allow for identification of genotypes that consequently produce high contents of phenolics in various weather conditions. If not done, this should be carefully discussed and underlined in the manuscript, as it is risky drawing strong conclusions based on a 1 year field trial.

The text “All wheat varieties (Table 1) were ground with a laboratory grinder (CemotecTM, 166 CM290, Denmark) for two minutes.” Should be revised (grain was ground, and not the varieties…).

Very small SD values in Tables 2 and 5 may indicate that these values show variation between 3 laboratory (technical) replicates and not the 2 field trial replications (even though it is mentioned in the methodology that “Field trials utilized a randomized complete block design with two replicates”). This should be clarified. The SD values should reflect variation between the field replications of the trial, and not technical, laboratory replications of the analysis (and also the raw data from each of the field replicates, and not laboratory technical replicates, should be included in the dataset undergoing statistical tests).

Under tables 3 and 4 there is a statement “Values with different letters in each wheat sample and fraction indicate statistically significant differences at p < 0.05.”, while in fact there are no letters in the tables.

Regarding: “Supplementary Materials: The following supporting information can be downloaded at: www.mdpi.com/xxx/s1, Figure S1: title; Table S1: title; Video S1: title.” – if there are no Supplementary Materials, this section should be deleted.

Regarding “Conflicts of Interest: Declare conflicts of interest or state “The authors declare no conflict of interest.” “ – this section needs to be completed following the journal requirements.

Regarding Funding an Acknowledgments sections: There is no need to repeat the same information in the Funding and Acknowledgments section. The currently provided text should be written in Funding section only.

The References do not follow the style of the Foods journal and should all be revised accordingly.

Author Response

Comments and Suggestions for Authors
-1

The manuscript is generally well-written and touches on an interesting subject of antioxidant activity and phenolic compounds’ profiles and contents in different genotypes of purple wheat. However, there are some aspects that need to be further explained and/or corrected.

S.1: The manuscript title should be revised/corrected (incorrect: “Total Phenolic Compound”; “Phenolic Acid Profile of Phenolic Compounds”). Suggestion: Antioxidant Activity and Profiles of Phenolic Compounds in Various Genotypes of Purple Wheat.

A.1: Thank you for your suggestion. We revised the title as “Antioxidant Activity and Profiles of Phenolic Acid in Various Genotypes of Purple Wheat”.

S.2: In the Introduction section it is mentioned (and repeated a couple of times) that the primary and most prevalent phenolic acid in wheat grain (also in purple wheat) is ferulic acid, while the presented study indicates other acids as predominant in the tested samples (gallic acid and protocatechuic acid in free and bound fractions of wheat samples, respectively). This should be discussed and explained.

A.2: The breeding process applied in our study may have caused differentiation in the phenolic contents of wheat. Because the species with low ferulic acid content may have been dominant in the hybridization process. In addition, the extraction method and climatic conditions may also have affected this result. The following sentences have been added to the Discussion section

It was reported from previously mentioned published studies that the ferrulic acid content is major phenolic content in especially bound fraction. The reason for the difference in our study may be due to the extraction method, climatic conditions and breeding process applied in our study. It should also be taken into account that our study has only one year data.

S.3: According to the description of the methods, field replications were included in the study design, but another important factor – season (=weather conditions) – was not taken into account. The study was carried out for one year only. It is well known that season-to-season differences in the weather conditions can strongly impact crop composition, including the phenolic contents and profiles, thus the study should be carried out for at least two or three seasons to allow for the identification of genotypes that consequently produce high contents of phenolics in various weather conditions. If not done, this should be carefully discussed and underlined in the manuscript, as it is risky drawing strong conclusions based on a 1-year field trial.

A.3: The revision was performed. The wheats used in this study are new wheats obtained from the western Siberian region, developed using classical breeding methods. That's why we only have one-year data available. It is emphasized that there is only one-year data in the discussion section.

S.4: The text “All wheat varieties (Table 1) were ground with a laboratory grinder (CemotecTM, 166 CM290, Denmark) for two minutes.” Should be revised (grain was ground, and not the varieties…).

A.4: Thank you for your suggestion. We revised the sentence.

S.5: Very small SD values in Tables 2 and 5 may indicate that these values show variation between 3 laboratory (technical) replicates and not the 2 field trial replications (even though it is mentioned in the methodology that “Field trials utilized a randomized complete block design with two replicates”). This should be clarified. The SD values should reflect variation between the field replications of the trial, and not technical, laboratory replications of the analysis (and also the raw data from each of the field replicates, and not laboratory technical replicates, should be included in the dataset undergoing statistical tests).

A.5: Thank you for your suggestion. We revised the sentence.

S.6: Under tables 3 and 4 there is a statement “Values with different letters in each wheat sample and fraction indicate statistically significant differences at p < 0.05.”, while in fact there are no letters in the tables.

A.6: Thank you for your suggestion. We deleted the sentence.

S.7: Regarding: “Supplementary Materials: The following supporting information can be downloaded at: www.mdpi.com/xxx/s1, Figure S1: title; Table S1: title; Video S1: title.” – if there are no Supplementary Materials, this section should be deleted.

A.7: Thank you. We deleted the sentence.

S.8: Regarding “Conflicts of Interest: Declare conflicts of interest or state “The authors declare no conflict of interest.” “ – this section needs to be completed following the journal requirements.

A.8: Thank you for your suggestion. We revised the sentence.

S.9: Regarding Funding an Acknowledgments sections: There is no need to repeat the same information in the Funding and Acknowledgments section. The currently provided text should be written in Funding section only.

A.9: Thank you for your suggestion. We revised the sentence.

S.10: The References do not follow the style of the Foods journal and should all be revised accordingly.

A.10: Thank you for your suggestion. We revised the References.

Reviewer 2 Report

The work described in this manuscript is interesting and relevant given the importance of wheat in human diet and the high production of wheat in Russia.  The work is simple, as the authors only focused in the content of phenolic acids. I missed the content of other groups of phenolic compounds, which show higher antioxidant properties and bioactivity than phenolic acids. However, the study is very interesting due to the wheat cultivars that were analyzed.  These are my specific comments.

-In my opinion the title should be simplified. Bound phenols are generally insoluble until they are released by acid or alkaline hydrolysis. So, the words “bound” and “insoluble” are, in my opinion, reiterative in the title. Same thing with the words “free” and “soluble”. By the way, the term “Total phenolic compounds” might be deleted of title as the sum of free and bound phenols results in the content of total phenolic compounds.

-Please, indicate in the introduction if the phenolic composition of the tested cultivars has previously studies by others.

 -The isomeric conformations (o-, p-, m-, etc.) are generally written in italic letters.  

-The authors indicate that they evaluated the antioxidant activity. In my opinion, the authors evaluated the antioxidant capacity. The authors should study the differences between the antioxidant activity and antioxidant capacity and correct the manuscript accordingly.

-I suggest to replace “It is one of the most important sources of nourishment for humans” with “ It is one of the most important food for humans”.

-I did not understand where the generation number is (Table 1). Please, clarify. I can easily identify the cultivar/line and grain color in table 1.

- Please, review the numbering of titles: Please, replace “2.2. .”, “2.3. .“, “2.3.1..”, “2.3.2..”, “3.1. .”, “3.2. .” and “3.3. .”    with “2.2.”, “2.3.“, “2.3.1.”, “2.3.2.”,  and “3.1.”, “3.2.” and “3.3.”,   respectively.

-Please, explain what a “crude filter paper” is (page 5).

-Please, replace “falcon tube” with   tube”. Falcon is a brand. You might indicate that a conical tube was used.  

-The numbers in “Na2CO3”should be indicated as subscripts (page 6).

-Please, replace “optical density” with “absorbance” in page 6.

-Perhaps “contents” should be replaced with “content” in line 266.

-The redaction/presentation of the manuscript should be reviewed, including the English presentation. These are some examples where the reviewing of redaction/presentation is required:

+Please, review the following sentence: “The bound phenolic compounds have important functions such as high antioxidant properties and preventing the oxidation of bioactive compounds in the colon.” This sentence contains two ideas, which are reiterative.

+Please, review the redaction of this sentence: “Phenolic acids have lots of beneficial effect to human health due to their high antioxidants capacities and prevent the damage of cells by preventing oxidation reactions via scavenging free-radical”. It might be “Phenolic acids have many beneficial effects on human health due to their high antioxidant properties.”

+Please, review the redaction of the following sentence: “Gallic acid, benzoic acid derivatives and predominant phenolics in cereals, was the predominant phenolic acid found in purple wheat samples and was found in the free fractions”.

-Please review the following: “…dark blue grained wheat (Triticum aestivum L. cv. Hedong Wumai) and Swiss red wheat grain…” Perhaps “dark blue (Triticum aestivum L. cv. Hedong Wumai) and Swiss red wheat grains”.

Author Response

Comments and Suggestions for Authors
-2

The work described in this manuscript is interesting and relevant given the importance of wheat in human diet and the high production of wheat in Russia.  The work is simple, as the authors only focused in the content of phenolic acids. I missed the content of other groups of phenolic compounds, which show higher antioxidant properties and bioactivity than phenolic acids. However, the study is very interesting due to the wheat cultivars that were analyzed.  These are my specific comments.

S.1: In my opinion, the title should be simplified. Bound phenols are generally insoluble until they are released by acid or alkaline hydrolysis. So, the words “bound” and “insoluble” are, in my opinion, reiterative in the title. Same thing with the words “free” and “soluble”. By the way, the term “Total phenolic compounds” might be deleted of title as the sum of free and bound phenols results in the content of total phenolic compounds.

A.1: Thank you for your suggestion. We revised the title as “Antioxidant Activity and Profiles of Phenolic Acid in Various Genotypes of Purple Wheat”.

S.2: Please, indicate in the introduction if the phenolic composition of the tested cultivars has been previously studied by others.

A.2: The wheat samples used in this study are new varieties and advanced lines developed by classical plant breeding methods in western Siberia. It has not been encountered any studies on the phenolic composition of these or associated wheat samples in the related literature

 S.3: The isomeric conformations (o-, p-, m-, etc.) are generally written in italic letters.  

A.3: Thank you for your suggestion. We revised the isomeric conformations.

S.4: the authors indicate that they evaluated the antioxidant activity. in my opinion, the authors evaluated the antioxidant capacity. the authors should study the differences between the antioxidant activity and antioxidant capacity and correct the manuscript accordingly.

A.4: Thank you for your suggestion. We revised.

S.5: I suggest to replace “It is one of the most important sources of nourishment for humans” with “ It is one of the most important food for humans”.

A.5: Thank you for your suggestion. We revised the sentence.

S.6: I did not understand where the generation number is (Table 1). Please, clarify. I can easily identify the cultivar/line and grain color in table 1.

A.6: In this study, new species and advanced lines were obtained by applying the classical breeding method. That's why some varieties are not yet at the cultuvar level, but at the advanced line level. That's why it's not named in the table as cultivar.

S.7: Please, review the numbering of titles: Please, replace “2.2. .”, “2.3. .“, “2.3.1..”, “2.3.2..”, “3.1. .”, “3.2. .” and “3.3. .”    with “2.2.”, “2.3.“, “2.3.1.”, “2.3.2.”,  and “3.1.”, “3.2.” and “3.3.”,   respectively.

A.7: Thank you. We revised the numbering of titles.

S.8: Please, explain what a “crude filter paper” is (page 5).

A.8: Thank you for noticing. I made a typo. We deleted “crude”.

S.9: Please, replace “falcon tube” with   “tube”. Falcon is a brand. You might indicate that a conical tube was used.  

A.9: Thank you. We revised. We deleted “falcon.

S.10: The numbers in “Na2CO3”should be indicated as subscripts (page 6).

A.10: Thank you. We revised.

S.11: Please, replace “optical density” with “absorbance” in page 6.

A.11: Thank you. We replaced “optical density” with “absorbance”.

S.12: Perhaps “contents” should be replaced with “content” in line 266.

A.12: Thank you for your suggestion. We revised.

S.13: The redaction/presentation of the manuscript should be reviewed, including the English presentation. These are some examples where the reviewing of redaction/presentation is required:

- Please, review the following sentence: “The bound phenolic compounds have important functions such as high antioxidant properties and preventing the oxidation of bioactive compounds in the colon.” This sentence contains two ideas, which are reiterative.

-Please, review the redaction of this sentence: “Phenolic acids have lots of beneficial effect to human health due to their high antioxidant capacities and prevent the damage of cells by preventing oxidation reactions via scavenging free-radical”. It might be that “Phenolic acids have many beneficial effects on human health due to their high antioxidant properties.”

- Please, review the redaction of the following sentence: “Gallic acid, benzoic acid derivatives and predominant phenolics in cereals, was the predominant phenolic acid found in purple wheat samples and was found in the free fractions”.

- Please review the following: “…dark blue grained wheat (Triticum aestivum L. cv. Hedong Wumai) and Swiss red wheat grain…” Perhaps “dark blue (Triticum aestivum L. cv. Hedong Wumai) and Swiss red wheat grains”.

A.13: Thank you. We revised the sentences.

Reviewer 3 Report

Dear Authors,

The present study evaluates the total phenolic content, phenolic compositions and antioxidant activity in the grain of 40 purple wheat samples. The research subject is interesting and brings scientific important data in the field, as it deals with a subject that is currently of great interest. Some changes of the manuscript should nevertheless be performed in order to improve its quality. Following specific changes should thus be performed:

 Minor changes

            Is there a specific name pf the purple cultivar or is it a variety of wheat? Please mention.

            All scientific names of species should be italic. Please change throughout the whole manuscript. “subsp.” does not need to be italic.

Major changes

Abstract: it should follow the structure of the manuscript – in the present form it only presents the purpose, results and a conclusion. Please mention methods that helped you get the presented results.

Introduction: This part should contain information regarding similar studies existing in scientific literature and, in comparison, authors should emphasize the novelty and originality of their study. Please add some explanations on what your study brings in novelty. It is absolutelty necessary. Similar studies must be cited. It is absolutely clear that the study is not well documented, as it has a reduced number of references. The purpose of the study needs to be rephrased to become clearer and needs to be found in the last paragraph. Please add further information and justifications and modify accordingly. Please offer a rationale for choosing this specific wheat cultivar/variety and make sure you clarify if it represents a cultivar or a variety. Make connections between all these aspects and the ones you present in this section and offer more details about them in the context of your study.

Discussions: The whole section does not discuss the results of your study, offering both connections between different concepts and explanations for your findings, as it should. Here you should emphasize novelty and originality of the present study once again. It is not mentioned in any place and it must be treated, as yor study treats a highly studied subject and it is absolutely necessary to mention what you bring in novelty. You need to compare your results with the ones obtained by other authors and you need to highlight what you bring in novelty compared to these. You need to offer more details and develop this part in order to bring consistency to your study. This should be performed both for the study of phenolic compounds and for the one of the biological activity.

Conclusions: Please offer perspectives of your study.

All these suggested changes should be performed in order to bring further improvements to the manuscript. 

Author Response

Comments and Suggestions for Authors
-3

Dear Authors,

The present study evaluates the total phenolic content, phenolic compositions and antioxidant activity in the grain of 40 purple wheat samples. The research subject is interesting and brings scientific important data in the field, as it deals with a subject that is currently of great interest. Some changes of the manuscript should nevertheless be performed in order to improve its quality. Following specific changes should thus be performed:

 Minor changes

S.1:  Is there a specific name of the purple cultivar or is it a variety of wheat? Please mention.

A.1: Wheat originating from West Siberia and grown in that region. These are new wheats developed with conventional plant breeding methods. Most of them are not cultivars yet and are technically called advanced line.

S.2: All scientific names of species should be italic. Please change throughout the whole manuscript. “subsp.” does not need to be italic.

A.2: The revision was performed. All scientific names was written in italic. 

Major changes

S.3: Abstract: it should follow the structure of the manuscript – in the present form it only presents the purpose, results, and a conclusion. Please mention methods that helped you get the presented results.

A.3: Thank you. We briefly mentioned the methods. “In this study, purple wheats were investigated in terms of their composition of free and bound phenolic acids and 2,2-diphenyl-1-picrylhydrazyl radical scavenging capacity.”

S.4: Introduction: This part should contain information regarding similar studies existing in scientific literature and, in comparison, authors should emphasize the novelty and originality of their study. Please add some explanations on what your study brings in novelty. It is absolutelty necessary. Similar studies must be cited. It is absolutely clear that the study is not well documented, as it has a reduced number of references. The purpose of the study needs to be rephrased to become clearer and needs to be found in the last paragraph. Please add further information and justifications and modify accordingly. Please offer a rationale for choosing this specific wheat cultivar/variety and make sure you clarify if it represents a cultivar or a variety. Make connections between all these aspects and the ones you present in this section and offer more details about them in the context of your study.

A.4:   The wheat used in this study was obtained from the western Siberia region. The wheats in this region were obtained by using classical breeding methods. In this respect, our work is quite original. In addition, some of these wheats are not yet at the cultivar level at the line level. Wheat varieties at the cultivar level are given in Table 1. There is no study on the phenolic contents of the new wheat species obtained. The following sentences were added to introduction section:

The wheat samples used in this study are new varieties and advanced lines developed by classical plant breeding methods in western Siberia. It has not been encountered any studies on the phenolic composition of these or associated wheat samples in the related literature.

S.5: Discussions: The whole section does not discuss the results of your study, offering both connections between different concepts and explanations for your findings, as it should. Here you should emphasize novelty and originality of the present study once again. It is not mentioned in any place and it must be treated, as yor study treats a highly studied subject and it is absolutely necessary to mention what you bring in novelty. You need to compare your results with the ones obtained by other authors and you need to highlight what you bring in novelty compared to these. You need to offer more details and develop this part in order to bring consistency to your study. This should be performed both for the study of phenolic compounds and for the one of the biological activity.

A.5: The wheat used in this study was obtained from the western Siberia region. The wheats in this region were obtained by using classical breeding methods. In this respect, our work is quite original. In addition, some of these wheats are not yet at the cultivar level at the advanced line level. Wheat varieties at the cultivar level are given in Table 1. There is no study on the phenolic contents of the new wheat species obtained. We discussed the results of previous studies on purple wheat varieties in the literature. However, since these wheat species were bred for the first time, there is no study on these varieties and advanced lines.

S.6: Conclusions: Please offer perspectives of your study.

A.6: Revision was performed. In this study, improved new wheat varieties were used. The breeding process may have caused a variation in the phenolic contents of the wheats. In the second stage of this study, cultivars and advanced lines with high characteristics in terms of values such as ferrulic acid content, antioxidant activity and total phenolic content will be considered.

All these suggested changes should be performed in order to bring further improvements to the manuscript.

Round 2

Reviewer 1 Report

Sadly some of my major comments were either very superficially addressed or completely ignored in this first round of revisions, in particular:

Re A.3: My comment was almost completely ignored and only one sentence was introduced into the discussion section, also in other context (in relation to ferulic acid only), which is not satisfactory. This significant weakness of this study should be strongly underlined and discussed in the discussion section and also mentioned in the final conclusions, which should be formed carefully, considering very often reported year-to-year variation in the phenolic contents of crops (even if grown in the same/very similar environment). The Authors should be able to convince the readers that this weakness does not undermine the relevance and importance of this study and its suitability for publication.

For my comment: S.5 („Very small SD values in Tables 2 and 5 may indicate that these values show variation between 3 laboratory (technical) replicates and not the 2 field trial replications (even though it is mentioned in the methodology that “Field trials utilized a randomized complete block design with two replicates”). This should be clarified. The SD values should reflect variation between the field replications of the trial, and not technical, laboratory replications of the analysis (and also the raw data from each of the field replicates, and not laboratory technical replicates, should be included in the dataset undergoing statistical tests”)) I received a reply: „A.5: Thank you for your suggestion. We revised the sentence.”. This does not answer the question, which was completely ignored by the Authors. And there is NOTHING new in the manuscript/no sentences revised regarding this problem. The Authors changed information about field replicated in the methods section (2 field replicates -> 3 field replicates), which makes it even more strange that the reported SDs were so small. What were the means and SDs in the tables based on?

Reg. A2 (lines 320-324 in the revised manuscript): If the Authors suppose that the extraction method applied probably did not allow to extract the main phenolic acid from the sample (or rather allowed to extract only partly), this totally undermines credibility of this method and the study outcomes… Moreover, English language correction of these newly introduced sentences needs to be applied.

Other, more minor aspects:

Reg. A1 (title): Acid -> Acids

Lines 93-94 in the revised manuscript: (“It has not been encountered any studies on the phenolic composition of these or associated wheat samples in the related literature.”): English language correction needs to be applied.

Lines 279-280 in the revised manuscript: Reference (citation) for this newly introduced sentence should be provided.

For my comment S.10 („The References do not follow the style of the Foods journal and should all be revised accordingly”) I received a reply: „A.10: Thank you for your suggestion. We revised the References.”. And I see that still full journal names are presented in the references (while there should be abbreviations of the journals’ names). This should be revised.

It is also important to point out that not all introduced changes have been clearly marked in the revised manuscript, and no clear reference of the introduced changes to the manuscript lines was given,  which makes it altogether more difficult to do a proper check of the introduced revisions.

Author Response

Comments:  My comment was almost completely ignored and only one sentence was introduced into the discussion section, also in other context (in relation to ferulic acid only), which is not satisfactory. This significant weakness of this study should be strongly underlined and discussed in the discussion section and also mentioned in the final conclusions, which should be formed carefully, considering very often reported year-to-year variation in the phenolic contents of crops (even if grown in the same/very similar environment). The Authors should be able to convince the readers that this weakness does not undermine the relevance and importance of this study and its suitability for publication.

Response: Thank you very much for your contribution. The aim of the our research was detailed in introduction section. The aims of our study and other previous studies on this subject are different. The aim of our study was to determine the lines with high antioxidant capacity and phenolic content by applying the breeding process. Thus, it is to obtain new species that can adapt to the Siberian region from new lines. In other studies, classical characterization of known species has been made.

It should also be taken into account that our study has only one year dataThis can be seen as a weakness for our study. However, in previously punlished studies, phenolic characterization of different known colored wheat cultivars was performed. In our study, unlike these studies, the effect of the breeding process on the amount and distribution of phenolic compounds in the new purple-grained lines was investigated. The results of our study suggested that new lines can adapt to the Siberian region with different phenolic content could be obtained by breeding process and could be considered as a new commercial cultivars.

Comments: For my comment: S.5 („Very small SD values in Tables 2 and 5 may indicate that these values show variation between 3 laboratory (technical) replicates and not the 2 field trial replications (even though it is mentioned in the methodology that “Field trials utilized a randomized complete block design with two replicates”). This should be clarified. The SD values should reflect variation between the field replications of the trial, and not technical, laboratory replications of the analysis (and also the raw data from each of the field replicates, and not laboratory technical replicates, should be included in the dataset undergoing statistical tests”)) I received a reply: „A.5: Thank you for your suggestion. We revised the sentence.”. This does not answer the question, which was completely ignored by the Authors. And there is NOTHING new in the manuscript/no sentences revised regarding this problem. The Authors changed information about field replicated in the methods section (2 field replicates -> 3 field replicates), which makes it even more strange that the reported SDs were so small. What were the means and SDs in the tables based on?

Response: Analyzes were made after the wheat from different parts of the field was blended. The sd value here shows the measurement differences obtained from laboratory analyzes. That's why the sd values are small.

Reg. A2 (lines 320-324 in the revised manuscript): If the Authors suppose that the extraction method applied probably did not allow to extract the main phenolic acid from the sample (or rather allowed to extract only partly), this totally undermines credibility of this method and the study outcomes… Moreover, English language correction of these newly introduced sentences needs to be applied.

Response: We do not have such an interpretation and preliminary studies have been made for the method we apply. In other words, after the method was completely settled, we continued our work with the final method. Therefore, we are confident that the extraction method is reliable. The language was checked and revised.

Other, more minor aspects:

Comments: Reg. A1 (title): Acid -> Acids

Response: The revision was performed.

Comments: Lines 93-94 in the revised manuscript: (“It has not been encountered any studies on the phenolic composition of these or associated wheat samples in the related literature.”): English language correction needs to be applied.

Response: The sentence was revised. (line 94-95)

Lines 279-280 in the revised manuscript: Reference (citation) for this newly introduced sentence should be provided.

Response: The references was cited (line 287-289)

Comments: For my comment S.10 („The References do not follow the style of the Foods journal and should all be revised accordingly”) I received a reply: „A.10: Thank you for your suggestion. We revised the References.”. And I see that still full journal names are presented in the references (while there should be abbreviations of the journals’ names). This should be revised. It is also important to point out that not all introduced changes have been clearly marked in the revised manuscript, and no clear reference of the introduced changes to the manuscript lines was given,  which makes it altogether more difficult to do a proper check of the introduced revisions.

Response: We used endnote style of mdpı journal. We also revised references. The abbreviation was used.   

Reviewer 3 Report

Dear Authors,

The present study evaluates the total phenolic content, phenolic compositions and antioxidant activity in the grain of 40 purple wheat samples. The authors performed most of the suggested changes after the first round of review. Following specific changes should still be performed:

 Major changes

Abstract: not all used methods are described. Please mention methods used for obtaining the results for the quantification of individual phenols.

Introduction: some explanations on what your study brings in novelty are nevertheless needed. Novelty is still not clear. The purpose of the study still needs to be rephrased to become clearer.

Discussions: I do not see that similar studies are cited, eventhough you sometimes describe their findings (e.g. line 320-321). Please highlight once more your novelty. It is important, as you deal with a highly studied species.

Conclusions: Please offer perspectives of your study.

All these suggested changes should be performed in order to bring further improvements to the manuscript.

Author Response

Dear Authors,

The present study evaluates the total phenolic content, phenolic compositions and antioxidant activity in the grain of 40 purple wheat samples. The authors performed most of the suggested changes after the first round of review. Following specific changes should still be performed:

 Major changes

Comments: Abstract: not all used methods are described. Please mention methods used for obtaining the results for the quantification of individual phenols.

Response: The HPLc method was desciribed for the quantification of individual phenols.

Comments: Introduction: some explanations on what your study brings in novelty are nevertheless needed. Comments: Novelty is still not clear. The purpose of the study still needs to be rephrased to become clearer.

Response: The aim of this research was detailed as below;

The wheat samples used in this study are new varieties and advanced lines developed by classical plant breeding methods in Western Siberia. We have not encountered any studies on the phenolic composition of these or associated wheat samples in the related literature. After these investigations promising purple-grained lines characterized by high antioxidant capacity and ferulic acid content are received which are candidates for selection of commercial cultivars adapted to the conditions of the West Siberian region and suitable for functional food production.

The aim of this study was to identify the best accessions on the phenolic contents with high antioxidant capacity for their further evaluation in the nurseries of breeding process, and the search for a marker for antioxidants and/or phenolic content in wheat. The variation in the grain color is quite wide (purple, dark purple, light brown, red, blue and black) and further investigations on the material by the co-authors are expected to reveal marker(s) for these significant grain characteristics.

Comments: Discussions: I do not see that similar studies are cited, eventhough you sometimes describe their findings (e.g. line 320-321). Please highlight once more your novelty. It is important, as you deal with a highly studied species.

Response: The difference of our manuscript was described in introduction section and discussion. Following part was added to discussion section;

However, in previously punlished studies, phenolic characterization of different known colored wheat cultivars was performed. In our study, unlike these studies, the effect of the breeding process on the amount and distribution of phenolic compounds in the new purple-grained lines was investigated. The results of our study suggested that new lines can adapt to the Siberian region with different phenolic content could be obtained by breeding process and could be considered as a new commercial cultivars.

Comments: Conclusions: Please offer perspectives of your study. All these suggested changes should be performed in order to bring further improvements to the manuscript.

Response: The conclusion part was improved as below;

The perspectives of this study demonstrate its successful application for development of purple-grained varieties of spring bread wheat, adapted to conditions of the West Siberian region, having high antioxidant capacity, and finally, suitable for functional food production. Further investigations on the material are expected to reveal marker(s) for these important grain characteristics.
